# Genetic architecture of band neutrophil fraction in Iceland

Gudjon R. Oskarsson [1,2], Magnus K. Magnusson [1,2 ✉], Asmundur Oddsson [1], Brynjar O. Jensson [1], Run Fridriksdottir[1], Gudny A. Arnadottir [1], Hildigunnur Katrinardottir[1], Solvi Rognvaldsson[1], Gisli H. Halldorsson [1], Gardar Sveinbjornsson[1], Erna V. Ivarsdottir [1], Lilja Stefansdottir[1], Egil Ferkingstad [1], Kristjan Norland[1], Vinicius Tragante[1,3], Jona Saemundsdottir[1], Aslaug Jonasdottir[1], Adalbjorg Jonasdottir[1], Svanhvit Sigurjonsdottir[1], Karen O. Petursdottir[1], Olafur B. Davidsson[1], Thorunn Rafnar [1], Hilma Holm [1], Isleifur Olafsson[4], Pall T. Onundarson[2,5], Brynjar Vidarsson[5,6], Olof Sigurdardottir[7], Gisli Masson[1], Daniel F. Gudbjartsson [1,8], Ingileif Jonsdottir [1,2,9], Gudmundur L. Norddahl[1], Unnur Thorsteinsdottir[1,2], Patrick Sulem [1] & Kari Stefansson [1,2 ✉]

The characteristic lobulated nuclear morphology of granulocytes is partially determined by composition of nuclear envelope proteins. Abnormal nuclear morphology is primarily observed as an increased number of hypolobulated immature neutrophils, called band cells, during infection or in rare envelopathies like Pelger-Huët anomaly. To search for sequence variants affecting nuclear morphology of granulocytes, we performed a genome-wide association study using band neutrophil fraction from 88,101 Icelanders. We describe 13 sequence variants affecting band neutrophil fraction at nine loci. Five of the variants are at the Lamin B receptor (*LBR*) locus, encoding an inner nuclear membrane protein. Mutations in *LBR* are linked to Pelger-Huët anomaly. In addition, we identify cosegregation of a rare stop-gain sequence variant in *LBR* and Pelger Huët anomaly in an Icelandic eight generation pedigree, initially reported in 1963. Two of the other loci include genes which, like *LBR*, play a role in the nuclear membrane function and integrity. These GWAS results highlight the role proteins of the inner nuclear membrane have as important for neutrophil nuclear morphology.

[1] deCODE genetics/Amgen Inc., Reykjavik, Iceland. [2] Faculty of Medicine, School of Health Sciences, University of Iceland, Reykjavik, Iceland. [3] Department of Cardiology, Division Heart and Lungs, University Medical Center Utrecht, Utrecht University, Utrecht, The Netherlands. [4] Department of Clinical Biochemistry, Landspitali, The National University Hospital of Iceland, Reykjavik, Iceland. [5] Department of Laboratory Hematology, Landspitali, The National University Hospital of Iceland, Reykjavik, Iceland. [6] The Laboratory in Mjodd, RAM, Reykjavik, Iceland. [7] Department of Clinical Biochemistry, Akureyri Hospital, Akureyri, Iceland. [8] School of Engineering and Natural Sciences, University of Iceland, Reykjavik, Iceland. [9] Department of Immunology of Landspitali, The National University Hospital of Iceland, Reykjavik, Iceland. ✉email: magnus.magnusson@decode.is; kstefans@decode.is

Granulocytes, also known as polymorphonuclear leuko-cytes, are white blood cells (WBC) characterized by a lobulated nucleus (2–5 lobes)[1,2]. The lobulated nucleus allows rapid extravasation and migration into interstitial tissue spaces during infection[3,4]. The nuclear morphology is partially determined by composition of nuclear envelope proteins[4,5]. Alterations in granulocytes' nuclear morphology are most commonly observed as an increased number of band neutrophils (and meta-myelocytes). Band neutrophils are immature neutrophils characterized by a hypolobulated and elongated nucleus and are routinely determined as part of differential blood count[6]. The abnormalities in nuclear morphology are seen in conditions such as infections[7] and myelodysplasia[8], or more rarely in congenital nuclear envelopathies such as familial Pelger-Huët anomaly[9].

Mutations in LBR have been found to segregate with Pelger-Huët anomaly in an autosomal dominant fashion in linkage studies[10–13]. Pelger-Huët anomaly is a rare, relatively benign, nuclear envelopathy characterized by functionally normal blood granulocytes with abnormal nuclear shape and chromatin organization[9,10]. In addition, homozygous loss-of-function of LBR is linked with Greenberg dysplasia[14–17], a prenatally lethal skeletal dysplasia[18], but is also reported in milder cases of skeletal abnormalities and non-lobulated granulocytes[10,19]. LBR encodes the lamin B receptor, a bifunctional protein localized on the nuclear inner membrane involved in the integrity of granulocyte nuclear shape through binding to lamin B proteins and hetero-chromatin and in sterol metabolism[10,15]. Even though Pelger-Huët anomaly affects nuclear morphology of granulocytes, it has not been established that the measurement of band neutrophil fraction can be used to identify Pelger-Huët anomaly[20].

In order to gain insight into sequence variants regulating neutrophil nuclear morphology, we performed a genome-wide association study (GWAS) on band neutrophil fractions. There are no reported associations of sequence variants with band neutrophil fraction, and this phenotype is not tested in available public datasets (i.e., UKB, BBJ, Finngen). We identified 13 sequence variants associated with band neutrophil fraction at nine loci, thereof five at the LBR locus. The variants at the LBR locus are of different frequencies (rare, low frequency, and common) and represent the primary genetic determinant of band neutrophil fractions. Subsequently, we unravel the genetic cause of Pelger-Huët anomaly in a previously described Icelandic pedigree by identifying an ultra-rare stop-gain variant, p.Arg76Ter, in LBR[21,22].

## Results

We performed a GWAS on 88,101 Icelanders with at least one assessment of band neutrophil fraction from complete blood count (CBC) (Supplementary Fig. 1 and Supplementary Table 1). Band neutrophil fraction is the percentage of WBC count made up of band neutrophils. The band neutrophil fraction used in the study was assessed from blood tests performed from 1993 to 2017, adjusted for sex, age at measure, different laboratories, and transformed to a standard normal distribution. In the GWAS we tested 34.0 million variants identified through whole-genome sequencing (WGS) of 49,962 Icelanders. The genotypes of 166,281 chip-typed individuals were then estimated using long-range phasing (imputation info >0.8 and MAF > 0.01%). Furthermore, the genotype probabilities of 290,482 first and second-degree relatives of chip-typed individuals were computed[23]. Thus, 26.7% of chip-typed and 15.0% of first and second-degree relatives of chip-typed individuals had information on band neutrophil fraction. Associations were considered significant if the p value was below a weighted, Bonferroni corrected, genome-wide significance threshold based on variant annotation[24]. Heritability of band neutrophil fraction in the Icelandic population was estimated to be 0.10 (95% CI 0.08–0.12) and 0.14 (95% CI 0.12–0.17) using parent-offspring and sibling correlations, respectively (Supplementary Table 2).

In the GWAS of band neutrophil fraction, we identified 13 distinct genome-wide significant associations with variants at nine loci (Table 1 and Fig. 1). To determine whether associations at a locus are distinct, we performed conditional analyses. Five of the band neutrophil associated sequence variants are at the LBR locus, three of which are common (MAF > 5%), one low-frequency (1% < MAF < 5%) and one rare (MAF < 1%). We note that the most significant association throughout the whole-genome is at LBR. Mutations in LBR have previously been linked with Pelger-Huët anomaly, a benign Mendelian condition characterized by morphological changes of granulocytes' nuclei[10]. Eight associations outside the LBR locus are represented by common variants (MAF ranging from 7.0–36.4%) with moderate effect sizes (Table 1). Out of the 13 band neutrophil fraction-associated variants, five are coding and one is a splice region variant. Band neutrophil fraction variance explained by all 13 identified variants is 1.50% and variance explained by the five variants at the LBR locus is 0.83%.

We tested the 13 band neutrophil fraction-associated variants for associations with ten hematological quantitative traits. We tested neutrophil nuclear segmentation and nine CBC phenotypes for all major lineages (granulocytes, neutrophils, basophils, eosinophils, monocytes, red blood cells, WBC, lymphocytes, and platelet count). Neutrophil nuclear segmentation is the average number of lobes neutrophils' nucleus have. This resulted in 130 tests (13 times 10) and we found 18 associations after adjusting for multiple testing (p value <0.05/130 = 3.8 × 10⁻⁴) (Table 2). Five variants were associated with neutrophil nuclear segmentation, four of which were at the LBR locus, and they all associated with opposite directions of effect to band neutrophil fraction (Table 2). However, no correlation was observed between the two phenotypes ($r^2 = 0.008$, $p = 0.018$). The three variants associated with quantitative hematological traits other than neutrophil nuclear segmentation were associated with WBC count and WBC lineages, and one of them was also associated with red blood cell count. We calculated genetic correlations between band neutrophil fraction in Iceland and eight quantitative blood traits from the UK Biobank using LD score regression (Supplementary Table 3)[25,26]. Band neutrophil fraction was found to correlate with neutrophil count ($r_g = 0.28$, SE = 0.065, p value = 0.028), but that association would not survive the number of tests performed for the different blood counts and no correlation was observed with the other blood counts. None of the band neutrophil fraction-associated variants were found to associate with diseases in the Icelandic population, given the number of phenotypes tested (significance threshold: $p < 0.05/441 = 1.1 \times 10^{-4}$). In order to search for genes and proteins mediating the effects of the sequence variants, we searched for a strong correlation ($r^2 > 0.80$) between the band neutrophil fraction-associated variants and variants that associate with transcript and protein levels in blood (based on 4730 plasma proteins measured in 35,559 Icelanders) and top cis-eQTL (based on RNA-sequencing analysis of blood from 13,175 Icelanders and a set of eQTL databases summary statistics). We found that four band neutrophil fraction-associated variants correlate with variants that affect levels of 11 plasma proteins (one in cis and 10 in trans) and seven variants correlating with variants affecting the levels of transcripts of 14 genes expressed in various tissues (Supplementary Table 4 and Supplementary Data 1 and 2).

**LBR locus**. At the LBR locus, we identified five sequence variants associated with band neutrophil fraction in ranges of frequencies

**Table 1 Associations of sequence variants under the additive model with normalized band neutrophil fraction in Iceland ($n = 88,101$).**

| Marker (hg38) | Amin/Amaj | MAF (%) | Info | LD size | Coding change | Coded gene | Other evidence in cis | Affected genes | Candidate gene | Effect (SD) [95% CI] | p value |
|---|---|---|---|---|---|---|---|---|---|---|---|
| rs724781 (chr1:153363542) | G/C | 26.7 | 1.00 | 7 | Downstream | - | eQTL, pQTL | S100A9 and S100A12 | S100A9 and S100A12 | -0.046 [-0.059; -0.033] | 6.6E-12 |
| rs2245425 (chr1:179889309) | G/A | 36.4 | 1.00 | 74 | Splice acceptor | TOR1AIP1 | sQTL | TOR1AIP1 | TOR1AIP1 | 0.033 [0.021; 0.045] | 6.9E-8 |
| rs41268715 (chr1:225333399) | C/G | 7.0 | 1.00 | 39 | p.Glu3232Gln | DNAH14 | - | - | LBR | -0.082 [-0.11; -0.057] | 9.6E-11 |
| rs17522489 (chr1:225346229) | G/A | 20.9 | 1.00 | 26 | p.Gln3556Arg | DNAH14 | - | - | LBR | 0.048 [0.033; 0.063] | 8.7E-11 |
| rs14205 (chr1:225401942) | G/A | 12.2 | 1.00 | 18 | 3' UTR | - | - | - | LBR | 0.13 [0.11; 0.15] | 2.5E-43 |
| rs80028106 (chr1:225402647) | T/C | 1.9 | 1.00 | 4 | 3' UTR | - | - | - | LBR | -0.15 [-0.19; -0.11] | 5.9E-11 |
| rs138769892 (chr1:225410316) | C/T | 0.33 | 1.00 | 33 | p.Tyr430Cys | LBR | - | - | LBR | 0.53 [0.43; 0.63] | 1.1E-26 |
| rs12613605 (chr2:43131771) | T/G | 24.0 | 1.00 | 53 | Regulatory region | - | - | - | - | 0.043 [0.030; 0.056] | 3.2E-10 |
| rs2036844 (chr5:126776461) | C/A | 27.6 | 1.00 | 21 | Upstream | - | eQTL | LMNB1 | LMNB1 | 0.052 [0.039; 0.065] | 2.3E-15 |
| rs2561758 (chr5:173778279) | A/G | 27.2 | 1.00 | 7 | Regulatory region | - | - | - | - | 0.047 [0.034; 0.060] | 6.8E-13 |
| rs757770077 (chr7:65987227) | Multi | 10.1 | 0.98 | 626 | Upstream | - | - | - | GUSB | -0.059 [-0.078; -0.040] | 2.2E-09 |
| rs7846314 (chr8:60738272) | T/A | 18.6 | 1.00 | 4 | Upstream | - | pQTL | - | CHD7 | 0.061 [0.046; 0.076] | 3.6E-16 |
| rs36084354 (chr19:1079960) | A/G | 14.1 | 1.00 | 3 | p.Met531Ile | HMHA1 | - | - | HMHA1 | 0.065 [0.048; 0.082] | 3.1E-14 |

Effect is shown for the minor allele in standard deviations.
*Amin* minor allele, *Amaj* major allele, *Coding change* the variant class annotation or the effects on coding sequence if applicable, *MAF* minor allele frequency, *Info* imputation information, *LD size* total number of variants correlating with $R^2 > 0.8$ to the variant, *eQTL* expression quantitative trait locus (QTL), *pQTL* protein QTL, *sQTL* splicing QTL, *SD* standard deviation.

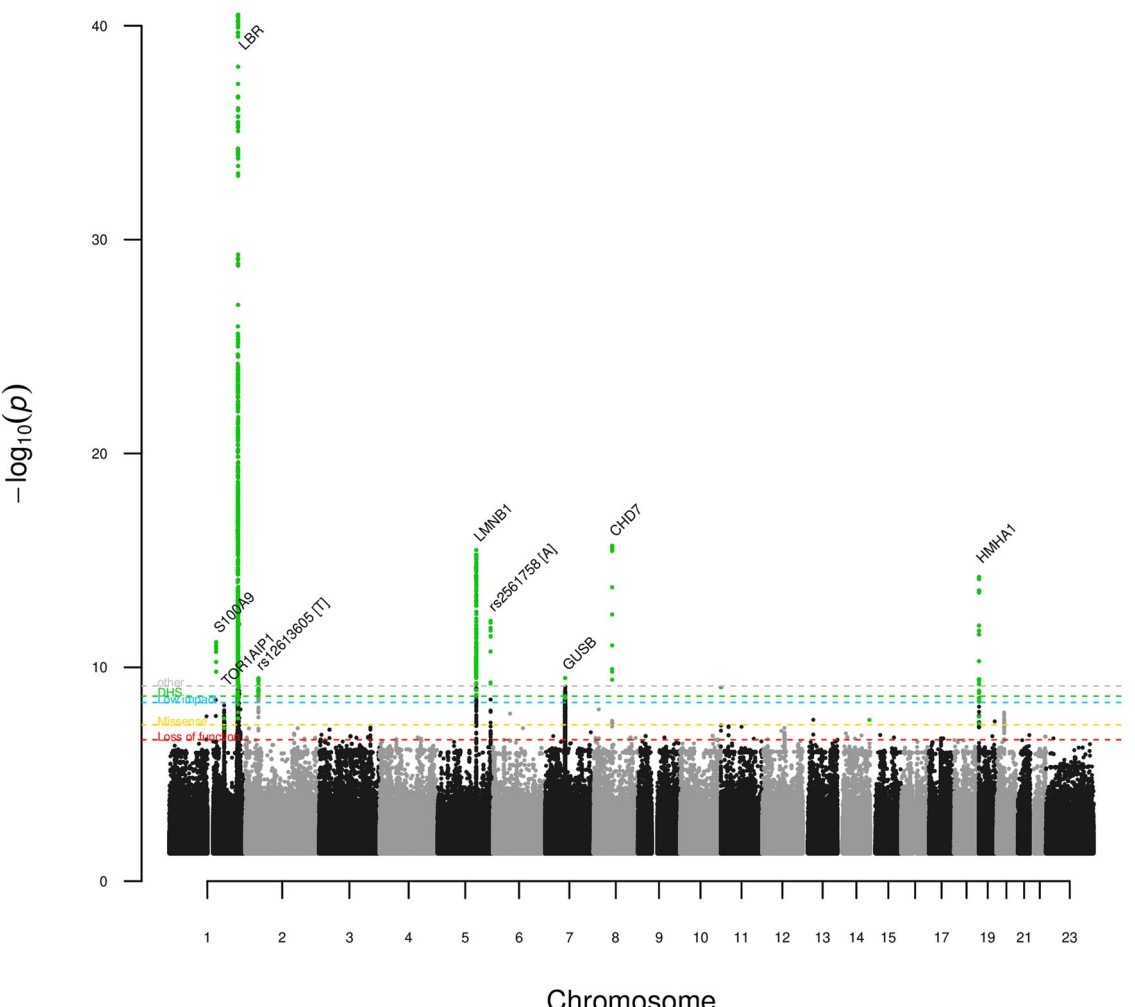

**Fig. 1 Manhattan plot for GWAS of band neutrophil fraction in the Icelanders ($N = 88,101$).** Variants at nine loci reached genome-wide significance and are labeled. All variants with $p$ values below their respective variant-class thresholds, indicated by horizontal dashed lines, are marked as green dots. Red dashed line: adjusted $p$ value significance threshold for variants predicted to lead to loss-of-function ($p$ threshold $= 2.4 \times 10^{-7}$). Yellow dashed line: adjusted $p$ value significance threshold for variants predicted to have moderate impact on gene function ($p$ threshold $= 4.9 \times 10^{-8}$). Cyan dashed line: adjusted $p$ value significance threshold for variants predicted to have low impact on gene function ($p$ threshold $= 4.4 \times 10^{-9}$). Green dashed line: adjusted $p$ value significance threshold for other variants in Dnase I hypersensitivity sites ($p$ threshold $= 2.2 \times 10^{-9}$). Gray dashed line: adjusted $p$ value significance threshold for all other variants ($p$ threshold $= 7.4 \times 10^{-10}$). Variants are plotted by chromosomal position ($x$ axis) and $-\log10[p]$ values ($y$ axis). For clarity, only variants with $p < 0.005$ are plotted.

and effect sizes, spanning a region of around 100 Kb. None of the five variants have been reported to associate with other traits in GWAS studies. All five variants remain significant after adjustment for the other four (Table 1 and Supplementary Tables 5 and 6).

The most significant of the five associations with band neutrophil fraction at *LBR* locus is with the common 3'-UTR variant rs14205[G], associating with higher band neutrophil fraction (MAF $= 12.2\%$; effect $= 0.13$ SD, $p = 2.5 \times 10^{-43}$) (Table 1 and Fig. 2). The 3'-UTR variant correlates highly with 17 variants at the locus ($r^2 > 0.80$). The two other common variants at the *LBR* locus associated with band neutrophil fractions, rs17522489[G] (MAF $= 20.9\%$) and rs41268715[C] (MAF $= 7.0\%$) (Effect $= 0.048$ SD, $p = 8.7 \times 10^{-11}$; Effect $= -0.082$ SD, $p = 9.6 \times 10^{-11}$), are both missense variants in *DNAH14*, which is in close proximity to *LBR* (<3 kb) (Table 1). Rs17522489[G] (p.Gln3556Arg) and rs41268715[C] (p.Glu3232Gln) are part of an LD class (variants $r^2 > 0.80$) of 26 and 39 variants, respectively. Considering the close proximity and linkage to *LBR* and the fact that variants in *LBR* are the cause of Pelger-Huët anomaly it is highly plausible that the

effects on band neutrophil fraction-associated with the variants in *DNAH14* are mediated through *LBR*.

The minor allele of the low-frequency 3'-UTR variant rs80028106[T] (MAF $= 1.9\%$) associates with lower band neutrophil fraction (Effect $= -0.15$ SD, $p = 5.9 \times 10^{-11}$) (Table 1). The variant with the largest effect on band neutrophil fraction is the rare missense variant NP_002287.2:p.Tyr430Cys (rs138769892[C]; MAF $= 0.33\%$; Effect $= 0.53$ SD, $p = 1.1 \times 10^{-26}$) (Table 1). One in 154 Icelanders are heterozygous carriers of p.Tyr430Cys and no homozygous carrier was observed, whereas around two are expected out of 166,281 imputed individuals (assuming Hardy-Weinberg equilibrium). The p.Tyr430Cys variant is observed in 394 of 220 K genomes (one homozygote) in gnomAD[27], primarily in Finnish and non-Finnish Europeans. The variant is located in exon ten, and the Tyr430 residue is located in the nuclear envelope inner membrane sterol reductase domain (Supplementary Fig. 2).

**Variants at other loci.** In addition to variants in *LBR*, we identified eight associations of sequence variants with band neutrophil fraction at other distinct loci, represented by common variants

**Table 2 Associations of the band neutrophil fraction-associated sequence variants and other relevant hematological quantitative phenotypes in the study.**

| Marker | Candidate gene | WBC | | Granulocytes | | Neutrophils | | Eosinophils | | Basophils | | Monocytes | | Lymphocytes | | RBC | | Platelet | | Neutrophil segmentation | |
|---|---|---|---|---|---|---|---|---|---|---|---|---|---|---|---|---|---|---|---|---|---|
| | | Effect (SD) | p | Effect (SD) | p | Effect (SD) | p | Effect (SD) | p | Effect (SD) | p | Effect (SD) | p | Effect (SD) | p | Effect (SD) | p | Effect (SD) | p | Effect (SD) | p |
| rs724781 (chr1:153363542) | S100A9 or S100A12 | 0.00 | 0.45 | 0.01 | 0.24 | 0.01 | 0.20 | −0.01 | 0.23 | −0.00 | 0.88 | −0.01 | 0.04 | 0.00 | 0.67 | −0.00 | 0.34 | −0.00 | 0.36 | 0.02 | 0.01 |
| rs2245425 (chr1:179889309) | TOR1AIP1 | −0.00 | 0.61 | −0.01 | 0.18 | −0.01 | 0.17 | 0.00 | 0.28 | 0.00 | 0.22 | −0.00 | 0.58 | 0.01 | 0.16 | −0.00 | 0.31 | 0.00 | 0.48 | −0.02 | 7.8E−3 |
| rs41268715 (chr1:225333399) | LBR | 0.01 | 0.53 | 0.01 | 0.24 | 0.01 | 0.16 | −0.01 | 0.52 | −0.01 | 0.06 | 0.00 | 0.93 | −0.01 | 0.37 | −0.00 | 0.83 | −0.00 | 0.82 | 0.06a | 7.0E−8a |
| rs17522489 (chr1:225346229) | LBR | 0.01 | 0.08 | 0.01 | 0.16 | 0.01 | 0.14 | 0.00 | 0.90 | 0.01 | 0.03 | 0.01 | 0.03 | 0.01 | 0.15 | 0.00 | 0.95 | 0.01 | 0.01 | −0.01 | 0.07 |
| rs142105 (chr1:225401942) | LBR | 0.00 | 0.66 | −0.00 | 0.91 | 0.00 | 0.95 | −0.00 | 0.80 | 0.01 | 0.06 | 0.01 | 0.21 | 0.01 | 0.40 | −0.01 | 0.28 | 0.00 | 0.71 | −0.05a | 2.8E−9a |
| rs8028106 (chr1:225402647) | LBR | 0.01 | 0.33 | 0.02 | 0.09 | 0.02 | 0.08 | −0.01 | 0.45 | −0.01 | 0.53 | 0.01 | 0.35 | 0.00 | 0.90 | 0.03 | 0.04 | 0.01 | 0.76 | 0.14a | 4.2E−11a |
| rs138769892 (chr1:225410316) | LBR | −0.02 | 0.62 | −0.01 | 0.67 | −0.01 | 0.68 | 0.02 | 0.66 | 0.00 | 0.99 | −0.06 | 0.09 | −0.05 | 0.16 | −0.05 | 0.13 | 0.01 | 0.86 | −0.25a | 8.2E−7a |
| rs12613605 (chr2:43131771) | - | −0.00 | 0.85 | −0.00 | 0.66 | −0.00 | 0.58 | 0.01 | 0.02 | 0.01 | 0.01 | 0.01 | 0.06 | −0.00 | 0.82 | 0.00 | 0.81 | 0.01 | 0.36 | −0.03a | 5.2E−6a |
| rs2036844 (chr5:126776461) | LMNB1 | −0.00 | 0.54 | −0.00 | 0.52 | −0.00 | 0.50 | −0.00 | 0.82 | 0.00 | 0.43 | 0.00 | 0.82 | −0.01 | 0.22 | −0.01 | 0.23 | −0.00 | 0.86 | −0.02 | 4.1E−4 |
| rs2561758 (chr5:173778279) | - | 0.02a | 7.5E−7a | 0.02a | 4.4E−7a | 0.02a | 2.4E−7a | 0.00 | 0.64 | 0.01 | 0.04 | 0.01 | 2.8E−3 | 0.01 | 0.02 | 0.01 | 0.05 | 0.00 | 0.82 | −0.01 | 0.44 |
| rs757770077 (chr7:65987227) | GUSB | 0.02 | 2.3E−3 | 0.02 | 3.6E−3 | 0.02 | 4.4E−3 | 0.01 | 0.48 | 0.00 | 0.59 | 0.01 | 0.12 | 0.00 | 1.00 | 0.00 | 0.87 | 0.00 | 0.60 | 0.03 | 2.1E−3 |
| rs7846314 (chr8:60738272) | CHD7 | 0.05a | 4.1E−18a | 0.05a | 4.1E−20a | 0.05a | 2.0E−22a | −0.02a | 6.1E−5a | 0.00 | 0.64 | 0.03a | 2.3E−6a | 0.01 | 0.05 | −0.02a | 1.5E−4a | −0.00 | 0.86 | 0.02 | 0.05 |
| rs36084354 (chr19:1079960) | HMHA1 | −0.03a | 1.9E−8a | −0.01 | 0.05 | −0.01 | 0.15 | −0.04a | 3.6E−11a | −0.01 | 0.02 | −0.02a | 3.2E−5a | −0.05a | 8.4E−16a | 0.01 | 0.18 | −0.02 | 2.8E−3 | −0.01 | 0.27 |

These are nine phenotypes that are part of the complete blood count (CBC) and one neutrophil morphological phenotype (neutrophil nuclear segmentation). Effect is shown in standard deviations for the minor allele.
WBC white blood cells, RBC red blood cells, SD standard deviation, p p value.
aAssociations reaching statistical significance after adjusting for multiple testing (p value <0.05/130 = 3.8 × 10−4) and effect sizes.

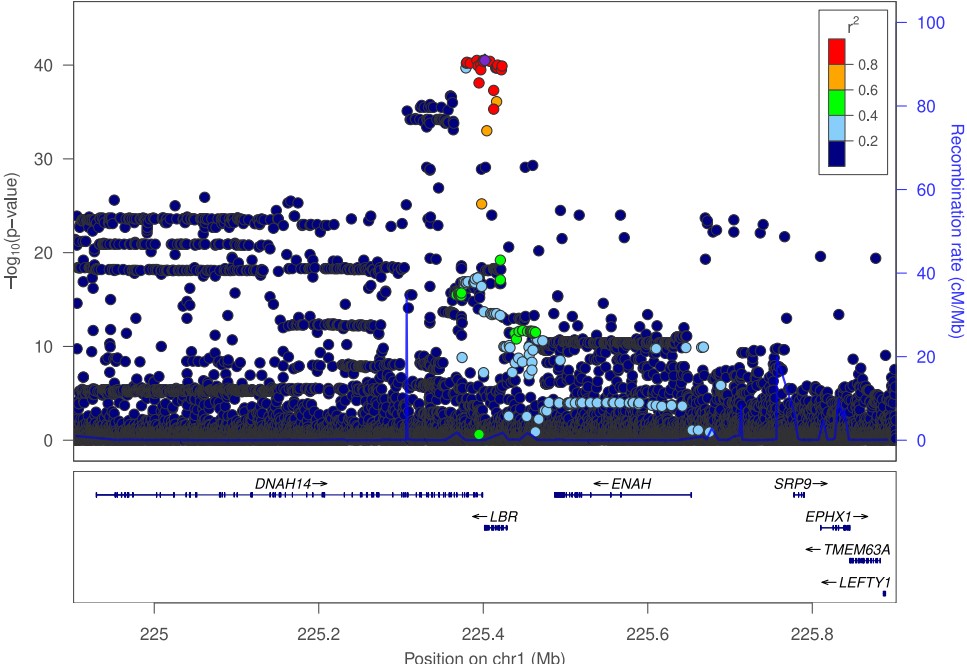

**Fig. 2 Associations of sequence variants at the *LBR* locus with band neutrophil fraction in Iceland.** Variants are colored according to correlation ($r^2$) to the most significant variant associated with band neutrophil fraction (legend at top-right). –log10p values along the left *y* axis and correspond to the variant depicted on the plot. The right *y* axis shows calculated recombination rates at the chromosomal location, plotted as a solid blue line. The common 3′-UTR variant rs14205[G] is the top marker associated with band neutrophil fraction and is depicted as a purple diamond.

with small effect (Table 1). Two of the variants are in genes encoding proteins interacting with the LBR on the inner membrane of the nuclear envelope (rs2036844[C] in *LMNB1*[28] and rs2245425[G] in *TOR1AIP1*[29]). Three of the eight sequence variants are also associated with WBC traits (rs2561758[A], rs7846314[T], rs36084354[A]) (Table 2).

The common upstream gene variant rs2036844[C] in *LMNB1* associates with higher band neutrophil fraction (MAF = 27.6%, effect = 0.052 SD, $p = 2.3 \times 10^{-15}$) (Table 1). *LMNB1* encodes lamin B1, a nuclear lamina component that provides a framework for the nuclear envelope and interacts with *LBR* and the chromatin[28]. In an RNA analysis of 13,175 Icelanders, the minor allele rs2036844[C] in *LMNB1* was found to be highly correlated ($r^2 = 0.97$) with the top cis-eQTL rs3014874[A], which is associated with lower levels of the *LMNB1* transcript (top cis-eQTL; effect = −0.49 SD, $p = 7.1 \times 10^{-271}$) (Supplementary Data 1 and 2). The allele associated with higher band neutrophil fraction associates with lower levels of the *LMNB1* transcript.

We also identified rs2245425[G], a splice-acceptor variant, in *TOR1AIP1* associating with increased band neutrophil fraction (MAF = 36.4%, effect = 0.033 SD, $p = 6.9 \times 10^{-8}$). *TOR1AIP1* encodes Torsin-1A-interacting protein 1, a protein required for nuclear membrane integrity and binds to A-and B-type lamins in the nuclear lamina[29]. We assessed effects of rs2245425[G] on splicing (sQTL) and found it to correlate with greater usage of an alternative acceptor site three bases upstream in *TOR1AIP1* (effect = 1.29 SD, $p < 1 \times 10^{-300}$) (Supplementary Figs. 3 and 4), which is predicted to maintain the reading frame and consequently to add an amino acid to the protein, thus making the consequence of this variant an inframe insertion.

Three common variants, rs2561758[A] on chromosome 5, rs7846314[T] at CHD7, and rs36084354[A] at HMHA1, are associated with increased band neutrophil fraction with effects ranging from 0.047 to 0.065 SD. All three variants have been reported to associate with other quantitative hematological traits, many of which we replicate[30,31] (Table 2). These results indicate

that the increased band neutrophil fraction observed for these markers is mediated through effects on hematopoietic differentiation. In a proteomic analysis, rs7846314[T] at *CHD7* was associated with plasma levels of six proteins in trans. Of interest, we notice that the level of one of the proteins encoded by *LCN2* is associated with two band neutrophil associated loci (*CHD7* and *S100A9*) (Supplementary Table 4). *LCN2* encodes lipocalin 2, also known as neutrophil gelatinase-associated lipocalin, which is a biomarker of human inflammatory diseases[32].

The minor allele of the common variant rs724781[G] in *S100A9* (MAF = 26.7%), associated with lower band neutrophil fraction (effect = −0.05 SD, $p = 7 \times 10^{-12}$) (Table 1), correlates strongly with top cis-eQTLs for lower RNA levels of *S100A9* and *S100A12* and cis-pQTL for lower serum protein levels of *S100A12* (Supplementary Tables 4–6). We are unable to distinguish whether the effect observed in the association of rs724781[G] with band neutrophil fraction is mediated through S100A9, S100A12, or both of them. However, we only identified significant pQTL for S100A12 and for the eQTL, the results for *S100A12* were more significant than for *S100A9*. Also, the V2G (variant to gene) scores, from Open Target Genetics, of rs724781[G] for *S100A9* and *S100A12* are 0.22 and 0.30, respectively, again favoring *S100A12*[33].

**LBR—familial Pelger-Huët anomaly**. Fewer loss-of-function variants in the *LBR* gene are observed than would be expected in large publicly available resources, such as gnomAD (LOEUF = 0.42)[27]. In Iceland, we observe only one predicted loss-of-function variant in *LBR*, a stop-gain variant (rs869312905[A], NP_002287.2:p.Arg76Ter), carried by two individuals out of 49,962 WGS. The p.Arg76Ter variant is absent from 220 K individuals in the gnomAD database[27]. This particular variant has been described in compound heterozygosity in a single case of anadysplasia-like bone dysplasia with Pelger-Huët anomaly[13]. It is located in the nucleoplasmic lamin B/DNA-binding domain and is predicted to lead to early termination in

**Table 3 A table of members of an Icelandic family with Pelger-Huët anomaly.**

| Individual | Pelger-Huët | Band neutrophil fraction | | Arg76Ter status |
|---|---|---|---|---|
| | | Standardized (SD) | Percentile | |
| IV.1 | Yes | – | – | Obligate |
| IV.4 | Unknown | – | – | Obligate |
| V.1 | Yes | – | – | Obligate |
| V.3 | Yes | – | – | Obligate |
| V.5 | Yes | – | – | Obligate |
| V.8 | Yes | – | – | Unknown |
| V.11 | Yes | – | – | Unknown |
| V.13 | Yes | – | – | Obligate |
| VI.2 | Yes | 2.40 | 99th | Carrier |
| VI.3 | Yes | 2.20 | 98th | Unknown |
| VI.5 | Yes | 1.30 | 91st | Carrier |
| VI.8 | Yes | – | – | Carrier |
| VI.9 | No | 2.10 | 98th | Unknown |
| VI.14 | No | 0.52 | 70th | Unknown |
| VI.16 | No | 1.30 | 91st | Unknown |
| VI.17 | No | 0.33 | 63rd | Unknown |
| VI.18 | Yes | – | – | Unknown |
| VI.19 | Yes | – | – | Unknown |
| VI.20 | Yes | – | – | Carrier |
| VI.21 | Yes | 1.00 | 85th | Carrier |
| VII.1 | Unknown | – | – | Carrier |
| VII.3 | Unknown | 1.60 | 94th | Carrier |
| VII.4 | Unknown | – | – | Carrier |
| VIII.1 | Unknown | 2.00 | 97th | Unknown |

A table of members of an Icelandic family with Pelger-Huët anomaly and their carrier status referring to pedigree shown in Fig. 3.
*Individual* shows the generation in Roman numbers and the individual number in Arabic number, *Pelger-Huët* diagnosis based on ref. [22], *SD* standard deviation.

the protein of 615 amino acids. We identified two closely related carriers of p.Arg76Ter in our the Icelandic whole-genome sequence, and through Sanger sequencing of 188 relatives, we identified six additional heterozygous carriers of this stop-gain variant. Based on genealogy data, ten obligate carriers were additionally detected (18 carriers in total). All the four p.Arg76-Ter carriers with assessed band neutrophil fraction (VI.2, VI.5, VI.21, VII.3) show large effect on the trait (from 1.0 to 2.4 SD) (Table 3). All the 18 identified p.Arg76Ter carriers belong to an eight-generation pedigree originating from ancestors born around 1775 in Austur-Skaftafellssýsla county in Southeast Iceland (Fig. 3). The p.Arg76Ter variant is absent outside of this family line. Two reports describing autosomal dominant Pelger-Huët anomaly in Iceland segregating within a multi-generation Icelandic pedigree ($N_{affected} = 15$) were published by Jensson et al. in 1963 and 1977, but genetic analysis was not available then. The pedigree described here (Fig. 3) is the same pedigree described in these previous reports. Of the 15 affected individuals included in the report from 1977, we confirmed ten as carriers or obligate carriers of the p.Arg76Ter and the other five were not analyzed since samples were not available (Fig. 3, Table 3 and Supplementary Table 7). Three additional carriers born after the publication of the initial reports could be identified in our data. Eleven out of 58 members of the pedigree have available band neutrophil fraction assessments (Table 3 and Supplementary Table 7). Genotypic data were available for four out of the 11 individuals with band neutrophil fraction, with all four individuals carrying the p.Arg76Ter variant. The mean of band neutrophil fraction before normalization is 3.4x higher in the four p.Arg76Ter carriers compared to 88,097 non-carriers (28.5% vs. 8.4%). Consistent with the previous report by Jensson et. al. from 1977, we find that a member of the pedigree (VI.3) reported to be affected with Pelger-Huët anomaly is an outlier for band neutrophil fraction (Fig. 3).

## Discussion

We performed a GWAS of band neutrophil fraction in the 88,101 Icelanders and detected 13 genome-wide significant associations represented by sequence variants at nine loci. Most notably, we describe five variants at the *LBR* locus influencing band neutrophil fraction, ranging from common variants with small effect sizes to rare coding variants with large effect sizes. We show that variation at *LBR* locus affects neutrophil morphology but not neutrophil count. This study also reveals that the *LBR* gene is the major genetic determinant of nuclear membrane morphology of granulocytes.

At the *LBR* locus on chromosome 1 (q42.12), three band neutrophil fraction-associated variants are in the LBR gene and two are missense variants in DNAH14 located immediately downstream of LBR. Mutations in *LBR* have been described in numerous studies to affect neutrophil nuclear morphology in cases of Pelger-Huët anomaly[10–13], whereas no variants in *DNAH14* are linked to the condition. Therefore, we speculate that the effects conferred by the five variants at the *LBR* locus are mediated through to effects on *LBR*.

We also identified two variants associated with band neutrophil fraction in *LMNB1* and *TOR1AIP1*, which encode inner nuclear membrane proteins of the nuclear lamina[28,34]. Intriguingly, *LMNB1* encodes the ligand for the Lamin B receptor, which is the gene product of *LBR*. The variants at *LMNB1* and at *TOR1AIP1* associate both with increased band neutrophil fraction and show effects on RNA. The altered function of genes encoding inner nuclear membrane proteins of the nuclear lamina is associated with effects on band neutrophil fraction, highlighting that the integrity of inner nuclear membrane proteins of the nuclear lamina is central to neutrophil nuclear membrane morphology.

Three variants in our results associate with other CBC phenotypes, primarily WBC phenotypes. The results indicate that higher band neutrophil fraction we observe for these variants is

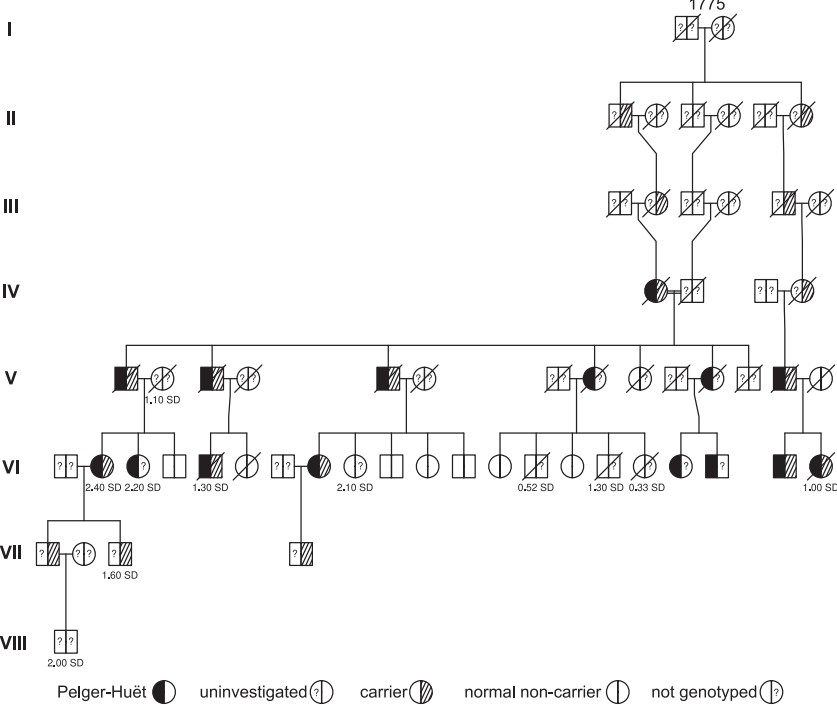

**Fig. 3 Pedigree of carriers of p.Arg76Ter in *LBR*.** All eight carriers (and ten obligate carriers) can be traced back to ancestors in the late eighteenth century. The founding couple have a total number of 9707 descendants. Roman numerals indicate generation, year of birth of the founding couple is noted above the symbols, and standardized band neutrophil fractions are noted below the symbols. Left part half-filled object = affected by Pelger-Huët anomaly according to ref. [22], question mark on left side = no data for the individual. We note that no genotypic data are available for the listed carriers in the loop formed in generations II, III, and IV. Their carrier status is based on the disease state of carrier IV.1. as presented in ref. [22].

due to changes in myeloid differentiation and/or proliferation. In these cases, the changes in band neutrophil fraction are likely due to increased count, but not morphological changes.

The Icelandic family affected by Pelger-Huët anomaly was described based on blood smear over 40 years ago[21,22]. Using genome sequencing methods, we identified the ultra-rare p.Arg76Ter cosegregating with and explaining the phenotype of the affected family members. Numerous pedigrees are available as a legacy from genetic epidemiology studies and from the era of linkage analysis, and the actual value of those past studies might lie in carefully curated multigenerational pedigrees[35]. Here, we demonstrate that when imputed genotypes based on whole-genome sequence data are available for a large part of a population, identification of rare deleterious variants within a well-defined pedigree.

Our results reveal that the band neutrophil fraction is substantially higher in cases of Pelger-Huët anomaly than in the general population, which suggests that Pelger-Huët anomaly can be detected from band neutrophil fraction assessment.

## Methods

**Study subjects**. The GWAS dataset used consisted of band neutrophil fraction assessments available from three different laboratories in Iceland from 88,101 Icelanders and obtained from 1993 to 2017. All participating individuals who donated blood, or their guardians, provided written informed consent. The family history of participants donating blood was incorporated into the study by including the phenotypes of first- and second-degree relatives and integrating over their possible genotypes. All sample identifiers were encrypted in accordance with the regulations of the Icelandic Data Protection Authority. Approval for the study was provided by the Icelandic National Bioethics Committee (ref: VSNb2015010033-03.12).

**Genotyping**. The study is based on testing 33,982,743 variants identified by WGS of 49,962 Icelanders, sequenced using Illumina standard TruSeq methodology to an average genome-wide coverage of 40X. SNPs and indels were identified and their genotypes called using joint calling with Graphtyper[36]. The effects of sequence

variants on protein-coding genes were annotated using the Variant Effect Predictor using protein-coding transcripts from RefSeq. We carried out chip-typing of 166,281 Icelanders (around 50% of the population) using Illumina SNP arrays. The chip-typed individuals were long-range phased[23], and the variants identified in the WGS of Icelanders were imputed into the chip-typed individuals (Imputation Info >0.8 and MAF > 0.01%). In addition, genotype probabilities for 290,482 untyped close relatives of chip-typed individuals were calculated based on Icelandic genealogy. The whole-genome sequenced samples were variant called jointly and the sequence variants found through WGS were phased jointly. The process used for whole-genome sequence sequencing of Icelanders, and the subsequent imputation from which the data for this analysis were generated has been extensively described in recent publications[37,38].

**Determination of the origin of sequence variants**. Close to complete genealogical records of the Icelandic population are available dating back to the Icelandic national census of 1703, and incomplete records dating back to the settlement of Iceland in 874 CE[39,40]. The Icelandic genealogy coupled with the large fraction of the population that has been chip-typed allows us to determine the origin of sequence variants through long-range phasing and haplotype imputation[37]. We used the Icelandic genealogy database[40,41] to identify the most recent common ancestors of carriers of the rare *LBR* sequence variant Arg76Ter, where all carriers shared a common ancestor. The sequence variant is absent from descendants of close relatives of the common ancestor carrying the same haplotype background.

**Phenotypes**. We used 258,312 band neutrophil fraction assessments from 88,101 Icelanders from three different laboratories in Iceland from 1993–2017 (Supplementary Table 1). Of the 88,101 individuals with band neutrophil fraction assessment, 44,438 were chip-typed and 43,663 were first- or second-degree relatives of chip-typed.

In the laboratories, band neutrophil fraction was measured and assessed using routine automated and semi-automated hematology analyzers. Band neutrophil fractions for each sex and the three different laboratories were separately transformed to a standard normal distribution and adjusted for age using a generalized additive model[42,43].

**Fraction of variance explained**. Heritability of band neutrophil fraction was estimated in the following two ways: (1) 2 × parent-offspring correlation, (2) 2 × full sibling correlation, using the Icelandic data (where all family relationships are known). The fraction of variance explained is calculated using the formula $2f(1 - f) a^2$ where $f$ is the minor allele frequency and $a$ is the additive effect[44].

Calculating the fraction of variance explained for variants in the GWAS catalog, we estimated the effects of published variants with corresponding phenotypes available in the deCODE data and calculated the fraction of variance explained using $f$ and $a$ obtained from the Icelandic population.

**Association analysis**. We performed a GWAS on 88,101 individuals from Iceland with at least one band neutrophil fraction assessment. Quantitative traits were tested using a linear mixed model implemented in BOLT-LMM[45]. We tested 33,982,743 variants (with imputation information >0.8 and MAF > 0.01%) identified from the WGS of 49,962 Icelanders (~16% of the population) for association with band neutrophil fraction. For binary phenotypes, sex, county of birth, current age or age at death (first and second order terms included), blood sample availability for the individual and an indicator function for the overlap of the lifetime of the individual with the time span of phenotype collection were included as covariates in the logistic regression model. The quantitative traits were transformed to a standard normal distribution.

For the study we used linkage disequilibrium (LD) score regression to account for distribution inflation in the dataset due to cryptic relatedness and population stratification[26]. Using a set of about 1.1 million sequence variants with available LD score, we regressed the $\chi^2$ statistics from our GWAS scan against LD score and used the intercept as correction factor. The estimated correction factor for band neutrophil fraction based on LD score regression was 0.91 for the additive model. When testing the association of sequence variants with quantitative traits, a BOLT linear mixed model was applied. These models are now widely used as they account for cryptic relatedness while also increasing power[45]. One-step in the BOLT-LMM procedure (step 1b) is to calibrate the $\chi^2$ test statistic by calculating a constant calibration factor. To compute the calibration constant BOLT-LMM rapidly computes the prospective statistic at 30 random SNPs by applying conjugate gradient iteration. However, this scaling was not applied to the test statistic in our association model. Therefore, when we applied the LD score regression and estimate a correction factor from the regressions intercept it was shifted by this constant factor. The correction factor can thus indeed be below one due to the calibration factor. The intercept is therefore not comparable to correction factors obtained from standard genomic control methods and should not be interpreted as such. Expected allele counts for sequence variants were used as covariates in the regression to test for association with other sequence variants conditional on their effects.

We calculated genetic correlations between Neutrophil bands and other traits as follows: we used cross-trait LD score regression and summary statistics for Neutrophil bands in the deCODE dataset and from the UKB dataset for other traits. In these analyses, we used results for about 1.2 million well imputed variants, and for LD information we used precomputed LD scores for European populations (downloaded from: https://data.broadinstitute.org/alkesgroup/LDSCORE/eur_w_ld_chr.tar.bz2).

**Conditional analysis**. We performed conditional analysis for each ±2 Mb region that contains at least one variant with genome-wide significant association with the studied trait in our data. All variants in the area with info >0.8 were included in the analysis with the lead variant (lowest $p$ value) as covariate. Variants were concluded to belong to an independent signal if their adjusted $p$ value was genome-wide significant. Conditional analysis was repeated for each region until a result with no genome-wide significant adjusted $p$ value was attained.

**Significance threshold**. We applied genome-wide significance thresholds corrected for multiple testing using adjusted Bonferroni procedure weighted for variant classes and predicted functional impact. With 33,982,743 sequence variants being tested in the Icelandic dataset, the weights given in Sveinbjornsson et al. were rescaled to control the family-wise error rate[22]. The adjusted significance thresholds are $2.4 \times 10^{-7}$ for variants with high impact ($N = 9147$), $4.9 \times 10^{-8}$ for variants with moderate impact ($N = 162,579$), $4.4 \times 10^{-9}$ for low-impact variants ($N = 2,361,837$), $2.2 \times 10^{-9}$ for other variants in Dnase I hypersensitivity sites ($N = 4,156,777$) and $7.4 \times 10^{-10}$ for all other variants ($N = 27,292,403$).

**Sanger sequencing**. Sequence variant G > A at chr1:225,422,217 (hg38), corresponding to p.Arg76Ter, was not imputed in our data due to the low frequency of the variant and low number of sequenced carriers in the original Icelandic dataset. A group of probable carriers of p.Arg76Ter and non-carriers were Sanger sequenced. Sanger sequencing confirmed 8 carriers of the p.Arg76Ter variant.

**RNA analysis**. Top cis-eQTL association results were collected from multiple publication and data sources listed in Supplementary Data 2[46–50] including GTEx[51] and deCODE cis-eQTL[52] resource with ($n = 13,175$) whole blood RNA-sequencing samples. We identified the top cis-eQTLs from all the eQTL data sources in deCODE sequence variant database and calculated genotypic correlation with all nearby variants to determine if any of them were in high LD ($r^2 > 0.80$) with the band neutrophil fraction GWAS variants. Gene expression was computed based on personalized transcript abundances estimated using kallisto[53]. Association between sequence variants and gene expression (cis-eQTL) were estimated using generalized linear regression, assuming additive genetic effect and quantile-normalized gene expression estimates, adjusting for measurements of sequencing artefacts, demography variables and 10 principal components of the expression matrix. To

investigate variants effects on splicing we utilized the 13,175 whole blood RNA-seq. samples. RNA-sequencing reads were aligned to personalized genomes using the STAR software package 13 with Ensembl v87[54,55] gene annotations. Quantification and association of alternative RNA splicing was done using LeafCutter[56]. To quantify the proportional usage of the two acceptor splice sites of exon 3 in *TOR1AIP1*, only fragments crossing the two splice sites were included. Samples with less than 5 total fragments crossing the two splice sites were removed.

**Plasma proteomics**. We tested the association of sequence variants with protein levels in plasma, measured using SOMAscan platform. We used an assay consisting of 5284 aptamers to estimate serum protein levels from 35,559 Icelandic individuals (34,818 chip-typed and 18,211 whole-genome sequenced) with genetic information and biological samples available at deCODE genetics[57]. Samples were collected between 2000 and 2019, with half coming from the Icelandic Cancer Project and the other half from various genetic projects at deCODE genetics, Reykjavik, Iceland. Samples' genotypic information was not made available to the staff running proteomics assays. Details of the method have been published elsewhere[58,59].

The assay provides measurement of relative binding of the plasma sample to each of the aptamers in relative fluorescence units (RFU). As a quality control, we calculated the correlation of log transformed RFU units over all 5284 aptamers for every pair of samples. We then calculated the average correlation of each sample with all other samples. The average correlation was high (median = 0.94) and samples with less than 0.82 correlation with other samples were excluded ($N = 421$). Furthermore, in order to evaluate the internal repeatability of the SOMAscan platform, we included 200 samples that were drawn from the same individuals at different time points and 228 that were replicated tubes of the same sample. In order to maintain consistency, we restricted the data to one sample per person, discarding all but the newest sample. For samples taken simultaneously as part of the same blood draw, one sample was selected at random. Finally, only 4983 aptamers measuring human proteins were assessed in the current analysis. These measure proteins encoded by a total of 4730 human genes.

**Reporting summary**. Further information on research design is available in the Nature Research Reporting Summary linked to this article.

## Data availability

Sequence variants passing GATK filters have been deposited in the European Variation Archive, accession number PRJEB15197. RNA-seq data have been deposited in the Gene Expression Omnibus, accession number GSE102870. The genome-wide association summary data are available at day of publication at http://www.decode.com/summarydata.

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

## Author contributions

G.R.O., M.K.M., B.O.J., R.F., D.F.G., U.T., P.S., and K.S. designed the study and interpreted the results. G.R.O., M.K.M., A.O., B.O.J., R.F., G.A.A., H.K., E.F., J.S., T.R., H.H., I.O., P.T.O., B.V., O.S., I.J., G.L.N., and P.S. carried out the subject ascertainment and recruitment. G.R.O., G.A.A., S.R., G.H.H., As.J., Ad.J., S.S., K.O.P., O.B.D., and P.S. performed the sequencing, genotyping, and expression analyses. G.R.O., M.K.M., A.O., B.O.J., R.F., G.A.A., H.K., S.R., G.H.H., G.S., E.V.I., L.S., E.F., K.N., V.T., S.S., K.O.P., O.B.D., G.M., D.F.G., and P.S. performed the statistical and bioinformatics analyses. G.R.O., M.K.M., D.F.G., U.T., P.S., and K.S. drafted the manuscript. All authors contributed to the final version of the paper.

## Competing interests

Authors affiliated with deCODE genetics/Amgen Inc., G.R.O., M.K.M., A.O., B.O.J., R.F., G.A.A., H.K., S.R., G.H.H., G.S., E.V.I., L.S., E.F., K.N., V.T., J.S., As.J., Ad.J., S.S., K.O.P., O.B.D., T.R., H.H., G.M., D.F.G., I.J., G.L.N., U.T., P.S., and K.S. declare competing interests as employees. The remaining authors declare no competing interests.
