## [Peer Review File · Communications Biology]

Reviewers' comments:

Reviewer #1 (Remarks to the Author):

In the manuscript submitted by Oskarsson and colleagues, the authors first performed GWAS on band neutrophil fractions in 88,101 Icelanders. Band neutrophil fraction is the fraction of band neutrophils (immature neutrophils characterized by a hypolobulated and elongated nucleus) in whole blood. There were 13 genome-wide significant leading SNPs from 9 loci (Table 1). The strongest association was observed in the LBR locus, which included 5 leading SNPs (the association signals were statistically independent). The LBR gene is biologically plausible for the target phenotype.

Next, the authors searched for loss-of-function variants of the LBR gene in 49,962 WGS of Icelanders. Only one predicted loss-of-function variant (rs869312905[A], NP_002287.2:p.Arg76Ter) was observed in two related individuals. By Sanger sequencing of the relatives, the authors discovered six additional carriers. The pedigree coincides with a previously reported pedigree with autosomal dominant Pelger-Huet anomaly (The old study did not perform DNA sequencing). By cosegregation, the authors determined that the p.Arg76Ter variant should be causal (Table 3, Figure 3).

I think the finding of the manuscript is valuable for deepening our knowledge on nuclear morphology of granulocytes. The study methodology is sound.

Figure 2

Genes other than LBR (such as DNAH14) should also be shown. The current figure gives false impression that the LBR exists in isolation.

Reviewer #2 (Remarks to the Author):

The authors provide a comprehensive study of band neutrophil fraction by using a GWAS and family pedigree analyses in the Icelandic population. The findings confirm the functional importance of LBR as previously reported in PH anomaly while highlighting novel rare and frequent variants at 9 loci. The study is robust and provides novel findings with regard to this phenotype. However, there are some points, which would need some clarifications.

As the study captured variants associated with the band neutrophil fraction in the Icelandic population, it is not clear whether some of those associations are also replicated in other populations.

Supp. Tables 3-4 report one cis pQTL (S100A9) and different eQTLs. As these snps may impact the phenotype, colocalization analysis may provide further insight as to whether these associations impacting the expression are possibly causal. Please, provide result of colocalization analyses between the GWAS and QTLs.

The present findings suggest that genes related with band neutrophil are shared with different blood cell phenotypes. However, it is not clear to which extent variants are shared. Cross-phenotype analysis including genetic correlation could provide further insights on this issue.

Please, provide the methodology for conditional and snp related variance analyses in the method section.

In the method section, please provide the pipeline for QTL analyses along with the different QTL resources that were used.

Reviewers' comments:

Reviewer #1 (Remarks to the Author):

In the manuscript submitted by Oskarsson and colleagues, the authors first performed GWAS on band neutrophil fractions in 88,101 Icelanders. Band neutrophil fraction is the fraction of band neutrophils (immature neutrophils characterized by a hypolobulated and elongated nucleus) in whole blood. There were 13 genome-wide significant leading SNPs from 9 loci (Table 1). The strongest association was observed in the LBR locus, which included 5 leading SNPs (the association signals were statistically independent). The LBR gene is biologically plausible for the target phenotype.

Next, the authors searched for loss-of-function variants of the LBR gene in 49,962 WGS of Icelanders. Only one predicted loss-of-function variant (rs869312905[A], NP_002287.2:p.Arg76Ter) was observed in two related individuals. By Sanger sequencing of the relatives, the authors discovered six additional carriers. The pedigree coincides with a previously reported pedigree with autosomal dominant Pelger-Huet anomaly (The old study did not perform DNA sequencing). By cosegregation, the authors determined that the p.Arg76Ter variant should be causal (Table 3, Figure 3).

I think the finding of the manuscript is valuable for deepening our knowledge on nuclear morphology of granulocytes. The study methodology is sound.

1. Figure 2

Genes other than LBR (such as DNAH14) should also be shown. The current figure gives false impression that the LBR exists in isolation.

Thank you for the observation. We have now added the neighboring genes at that locus to the figure.

Reviewer #2 (Remarks to the Author):

The authors provide a comprehensive study of band neutrophil fraction by using a GWAS and family pedigree analyses in the Icelandic population. The findings confirm the functional importance of LBR as previously reported in PH anomaly while highlighting novel rare and frequent variants at 9 loci. The study is robust and provides novel findings with regard to this phenotype. However, there are some points, which would need some clarifications.

1. As the study captured variants associated with the band neutrophil fraction in the Icelandic population, it is not clear whether some of those associations are also replicated in other populations.

There are no reported GWAS studies on band neutrophils when searching PubMed or Google scholar and the phenotype is not tested in publicly available sources such as UK Biobank, Biobank Japan and FinnGen (Finland). In addition, none of the international datasets we have access to, test for this phenotype.

We have now added a sentence, lines 64-66 page 3 now say:

“There are no reported associations of sequence variants with band neutrophil fraction, and this phenotype is not tested in available public datasets (i.e., UKB, BBJ, Finngen)”

2. Supp. Tables 3-4 report one cis pQTL (S100A9) and different eQTLs. As these snps may impact the phenotype, colocalization analysis may provide further insight as to whether these associations impacting the expression are possibly causal. Please, provide result of colocalization analyses between the GWAS and QTLs.

In the eQTL and pQTL data there is evidence for both S100A9 and S100A12 to be the mediator of the effect of rs724781 on band neutrophil fraction. We note that the pQTL and eQTL are more significant for S100A12 than S100A9. It can be conceived that the effect is mediated solely either by S100A12 or S100A9 and alternatively we cannot exclude a contribution from both genes. We currently cannot differentiate between these three scenarios.

For clarification, we have added a sentence in the results. Lines 185-187 page 7 now read:

“...RNA levels of S100A9 and S100A12 and cis-pQTL for lower serum protein levels of S100A12 (Supplementary Tables 4, 5, and 6). We are unable to distinguish whether the effect observed in the association of rs724781[G] with band neutrophil fraction is mediated through S100A9, S100A12, or both of them.”

3. The present findings suggest that genes related with band neutrophil are shared with different blood cell phenotypes. However, it is not clear to which extent variants are shared. Cross-phenotype analysis including genetic correlation could provide further insights on this issue.

We performed a genetic correlation analysis as suggested. Genetic correlation is shown in Supplementary Table X.

Supplementary Table X. Genetic correlation between band neutrophil fraction from Iceland and eight quantitative blood traits from the UK using LD score regression.

Secondary trait	Neutrophil bands		
	r_g	SE	P-value
Basophil count	0.13	0.082	0.10
Eosinophil count	0.038	0.067	0.58
Lymphocyte count	-0.063	0.061	0.31
Monocyte count	-0.060	0.056	0.29
Neutrophil count	0.14	0.065	0.028
Platelet count	0.011	0.043	0.79
Red blood cell count	0.061	0.059	0.30
White blood cell count	0.091	0.064	0.15

$r_g =$ genetic correlation, $SE =$ standard error

Text has now been added to the Results chapter of the manuscript and pages 4-5, lines 110-114 now say (changes highlighted):

“...and one of them was also associated with red blood cell count. We calculated genetic correlations between band neutrophil fraction in Iceland and eight quantitative blood traits from the UK Biobank using LD score regression (Supplementary Table 3)^{25,26}. Band neutrophil fraction was found to correlate with neutrophil count ($r_g = 0.28$, $SE = 0.065$, $P\text{-value} = 0.028$), but that association would not survive the number of test performed for the different blood counts and no correlation was observed with the other blood counts.”

Text has now been added to the Methods chapter of the manuscript and page 12, lines 329-333 now say (changes highlighted):

We calculated genetic correlations between Neutrophil bands and other traits as follows: We used cross-trait LD score regression and summary statistics for Neutrophil bands in the deCODE dataset and from the UKB dataset for other traits. In these analyses, we used results for about 1.2 million well imputed variants, and for LD information we used precomputed LD scores for European populations (downloaded from:

https://data.broadinstitute.org/alkesgroup/LDSCORE/eur_w_ld_chr.tar.bz2).

4. Please, provide the methodology for conditional and snp related variance analyses in the method section.

We have now added the following to a new section called Conditional analysis in the Methods clarifying the methodology for conditional analysis, page 12 lines 334-340:

“Conditional analysis

We performed conditional analysis for each ± 2 Mb region that contains at least one variant with genome-wide significant association with the studied trait in our data. All variants in the area with $\text{info} > 0.8$ were included in the analysis with the lead variant (lowest P-value) as covariate. Variants were concluded to belong to an independent signal if their adjusted P-value was genome-wide significant. Conditional analysis was repeated for each region until a result with no genome-wide significant adjusted P-value was attained.”

The title of the section called “Heritability” in the method section is now called “Fraction of variance explained”. We have also added text to the section.

The section now reads on page 11 lines 298-302 (changes highlighted):

“Heritability of band neutrophil fraction was estimated in the following two ways: 1) $2 \times$ parent-offspring correlation, 2) $2 \times$ full sibling correlation, using the Icelandic data (where all family relationships are known). The fraction of variance explained is calculated using the formula $2f(1-f)a^2$ where f is the minor allele frequency and a is the additive effect⁴³. Calculating the fraction of variance explained for variants in the GWAS catalog, we estimated the effects of published variants with corresponding phenotypes available in the deCODE data and calculated the fraction of variance explained using f and a obtained from the Icelandic population.”

5. In the method section, please provide the pipeline for QTL analyses along with the different QTL resources that were used.

We used eQTL results from multiple data sources that are listed in Supplementary Table 6 along with additional information. We have added reference into the manuscript to the most relevant publication of each data repository where the eQTL analysis pipeline is described. The RNA analysis method section did include detailed description of the RNA sequencing and eQTL association pipeline for the Icelandic RNA-seq data which has already been described in a recent publication [REF]. Since we are using all the eQTL data sources equally we feel there is no need to include more text on the Icelandic RNA-seq data. We removed the description of a method to detect imbalance allele specific expression which is no longer appropriate for the status of the manuscript.

The RNA analysis in the methods section now says, page 13 lines 355-369:

Top cis-eQTL association results were collected from multiple publication and data sources listed in Supplementary Table 6 including GTEx and deCODE cis-eQTL resource with (n = 13,175) whole blood RNA-sequencing samples. We identified the top cis-eQTLs from all the eQTL data sources in deCODE sequence variant database and calculated genotypic correlation with all nearby variants to determine if any of them were in high LD ($r^2 > 0.80$) with the band neutrophil fraction GWAS variants. Gene expression was computed based on personalized transcript abundances estimated using kallisto. Association between sequence variants and gene expression (cis-eQTL) were estimated using generalized linear regression, assuming additive genetic effect and quantile-normalized gene-expression estimates, adjusting for measurements of sequencing artefacts, demography variables and 10 principal components of the expression matrix. To investigate variants effects on splicing we utilized the 13,175 whole blood RNA-seq samples. RNA sequencing reads were aligned to personalized genomes using the STAR software package 13 with Ensembl v87 gene annotations. Quantification and association of alternative RNA splicing was done using LeafCutter. To quantify the proportional usage of the two acceptor splice sites of exon 3 in TOR1AIP1, only fragments crossing the two splice sites were included. Samples with less than 5 total fragments crossing the two splice sites were removed.

REVIEWERS' COMMENTS:

Reviewer #1 (Remarks to the Author):

The authors addressed my previous concern.

Reviewer #2 (Remarks to the Author):

The points that were raised during the review process have been addressed by the authors.
Thank you.